# In Search of the "Good Life": The Appeal of the Tiny House Lifestyle in the USA

**Severin Mangold** and **Toralf Zschau** *

Department of Sociology and Human Services, University of North Georgia, Dahlonega, GA 30597, USA;
mangolduk@gmail.com
* Correspondence: tzschau@ung.edu

**Abstract:** Over the past decade, tiny houses and the lifestyle they promote have become a world-wide phenomenon, with the trend especially impactful in the United States. Given their broad appeal and increasing prominence within popular culture, it is surprising how little research exists on them. To help to better understand what motivates people to adopt this lifestyle, this paper presents insights from an exploratory study in the United States and offers the first contours of a new conceptual framework. Situating the lifestyle within the larger economic and cultural forces of our times, it argues that going "tiny" is seen by tiny house enthusiasts as a practical roadmap to the *Good Life*: A simpler life characterized by more security, autonomy, relationships, and meaningful experiences. The paper ends with a brief discussion of broader implications and directions for future research.

**Keywords:** tiny house; lifestyle; movement

## 1. Introduction

While the idea of living simply—in the American context—can primarily be traced back to the writings of Henry David Thoreau, the modern tiny house movement takes hold with the writing of Lester Walker's (1987) book *Tiny, Tiny Houses or How to Get Away from It All* and Sara Susanka and Obolensky's (1998) *The Not So Big House: A Blueprint for the Way We Really Live* (Lasky 2016). With the hit of the 2008 global economic recession, many individuals found the idea of living in a tiny house[1] compelling, and traction within the tiny house (TH) movement began to pick up (Jones 2016). Much of what is known about the tiny house movement comes from major blogs (e.g., TinyHouseLife.com, tinyhouseblog.com), television shows (*Tiny House Nation*, *Tiny House Luxury*, *Tiny House Hunters*), newspaper and magazine articles, tiny house builders, and information disseminated during annual TH conferences and workshops. These sources paint the tiny house lifestyle as a rejection of the American *bigger-is-better* mantra, in which individuals seek a "better life" by downsizing belongings, becoming more conscious consumers, and lessening their environmental impact (Levin 2012; Nelson 2012; Buczynski 2014; Friedlander 2014; Hanks 2017; Jones 2016; Kronenburg 2017). In addition, tiny houses have been explored as solutions for homelessness (Lewis 2017), alternatives to nursing homes (Chaney 2017) and temporary shelters for disaster zones (Reggev 2017). Despite its growing popularity, however, the TH lifestyle has received little attention within academia (Mutter 2013; Boeckermann et al. 2017; Ford and Gomez-Lanier 2017). While tiny house living is a global phenomenon, this paper presents insights from an exploratory study in the United States of America.

---

[1] Tiny house enthusiasts usually make a distinction between small and tiny houses. True tiny houses, in this view, are dwellings (whether on foundation or wheels) smaller than 400 sqft. Small houses, by contrast, usually range from 400 sqft to 1000 sqft.

This research investigated the factors that make individuals adopt and/or consider adopting a TH lifestyle. Based on the analysis of thirty interviews with TH enthusiasts, the paper suggests that the tiny house lifestyle represents a contemporary answer to the age-old question: How does one live a Good Life?[2] The interviewees' answers, while multifaceted, converge on issues that center on desires to find security, autonomy, meaningful relationships, simple ways of living, and new experiences. The paper argues that while continuities with previous lifestyle movements exist, the TH lifestyle offers a much more individualistic, pragmatic, and experience-driven road to finding happiness. To situate the TH house phenomenon within the broader literature of similar lifestyle movements (Haenfler et al. 2012) and provide theoretical context for the study, the next section summarizes what draws individuals to similar lifestyle movements (primarily voluntary simplicity and minimalism).

## 2. Insights from Other Lifestyle Movements (LMS)

To our knowledge, no systematic studies on motivators for tiny house living in the United States exist. The tiny house movement has been influenced by and borrows heavily from a range of movements that share similar ideals, such as Thoreau's transcendentalism, the back-to-land movement, pragmatism, environmentalism, voluntary simplicity, downshifting, as well as minimalism (Nathan 2014; Kilman 2016; Heather 2017; Kamal 2017; Shearer and Burton 2018). Within the American experience, voluntary simplicity and minimalism, arguably more so than other movements, have helped to shape much of the tiny house movement's current vocabulary, cultural narratives, and its own self-understanding. By highlighting what draws individuals into the orbits of the lifestyles of these sister movements, the following section hopes to provide the conceptual contours needed to better map similar processes in the tiny house lifestyle.

### 2.1. Voluntary Simplicity (VS) Movement

Voluntary simplicity has a long history of influences from the Puritans, Quakers, and transcendentalism, and is often referred to as the "simple living" movement (Shi 1985). Much of the philosophical underpinnings of the movement were already articulated in *The Value of Voluntary Simplicity* (Gregg 2009). Later, Duane Elgin called VS a manner of living that is "outwardly more simple and inwardly more rich" (Elgin and Mitchell 1977, p. 2). Depending on the nature of people's ideological commitment, voluntary simplifiers can be grouped into "downshifters", "strong simplifiers", and "holistic simplifiers" (Etzioni 1998, pp. 110–13). Grigsby (2004, p. 2) maintains that VS aims to provide a blueprint to a "more fulfilling life" by "reduc[ing] clutter and minimizing activities" that one does not "find meaningful". In other words, VS can be seen as a "philosophy of living that advocates a counter-cultural position based on notions of sufficiency, frugality, moderation, restraint, localism, and mindfulness." (Alexander 2015, p. 8). However, people adopt VS for a variety of reasons: Interest in environmentalism, minimizing consumption, personal growth, and living closer with nature tend to be at the center of their narrative (Shama and Wisenblit 1984; Grigsby 2004; Alexander 2011). As Elgin (2006, p. 459) put it, "[t]he intention of voluntary simplicity is not to dogmatically live with less. It's a more demanding intention of living with balance." This may in part explain why individuals that adopt the VS lifestyle live happier lives and exhibit higher life satisfaction (Alexander and Ussher 2012; Boujbel and d'Astous 2012). Downshifting contained within VS according to Etzioni (1998) offers further lifestyle insights. Some have argued that downshifting

---

[2]　　Our views on the Good Life have been strongly influenced by Skidelsky and Skidelsky's 2012 treatise "How Much is Enough? Money and the Good Life". New York, Other Press LLC. The authors define the Good Life as "a life that is desirable, or worthy of desire". While specific conceptions have varied across time and space, Skidelsky and Skidelsky (2012) maintain that there is broad philosophical agreement on what constitutes the "basic goods [or elements]" of the Good Life (i.e., health, respect, security, relationships of trust and love). Building on Amartya Sen's theoretical work on human capabilities (e.g., "the capacity to be healthy"), the authors argue that these universal "basic goods . . . are not just means to, or capabilities for a good life; they are the good life."

is its own movement and that downshifters are people who choose to work fewer hours and spend less on consumerist goods, alleviating financial stress, which—in turn—gives them more time to nurture their relationships (Nelson et al. 2007; Chhetri et al. 2009; Kennedy et al. 2013). According to Chhetri et al. (2009, pp. 55–58), people express three main reasons to downshift, (1) "absconding from consumerism and stressful work", (2) "yearning to return to home life", and (3) "finding self and fulfilment". Others have introduced the distinction between "downshifters" and "simple-livers." Downshifters attempt to regain their financial independence by reducing their work hours and standard; simpler-livers, by contrast, will take more drastic changes (e.g., quitting their jobs) to achieve the same end (Schor 1999).

*2.2. Minimalism*

While most of what is known about minimalism comes from non-academic books, news articles, and blogs, these sources are informative. Minimalism has been described as a philosophy to intentionally eliminate excess things in order to live a simple life which places focus on items of importance and value (Wright 2010; Burnell 2017; Russell 2017). Rodriguez (2017) argues that the US minimalist movement is a mere individual response focused on reducing consumption; it lacks the radical potential to collectively challenge the capitalist system that often traps consumers in a primarily materialistic landscape[3]. Since minimalism means different things to different people, finding a clear and all-encompassing definition remains challenging. Furthermore, only a few movement insiders have attempted to articulate the minimalist philosophy. Leo Babauta's (2009) *The Simple Guide to a Minimalist Life* boils minimalism down to five core philosophical principles, which are meant to provide a simple how-to manual for individuals on the journey to a more meaningful life. Minimalists, to Babauta, are people who (1) omit needless things, (2) identify the essential, (3) make everything count, (4) fill their lives with joy, and (5) "edit, edit [*sic*]" (i.e., meaning engagement in this process is continuous). By elevating the quest for the essential to its experiential leitmotif, minimalism moves away from the more philosophical and spiritual underpinnings central to VS. While echoes of the eudemonic good life remain, contemporary notions of happiness with its focus on the visceral become much more central within minimalism. This subtle shift from the value-driven toward the experiential may help to explain why some authors argue that, rather than being a principled cultural rejection of conspicuous consumption, it actually represents a hidden embrace of it (Fagan 2017). Minimalism has received a lifeline with Joshua Fields Millburn and Ryan Nicodemus's 2014 TedTalk *Rich Life with Less Stuff* and their subsequent publication of *Minimalism: Live a Meaningful Life* (Millburn and Nicodemus 2015). In the book, the authors argue that minimalism is a "tool" to live "a meaningful life" because it helps "strip . . . away unnecessary things in . . . life so . . . [one] can focus on what's important" (Millburn and Nicodemus 2015, p. 25). People are attracted to the simple prescriptive elements of the movement precisely because they feel it liberates them from the struggles of contemporary life and allows them to chase their dreams (Jennifer 2015), travel (Amanda 2017), live in the moment (Altucher 2016), and seek true happiness (Becker 2015).

## 3. Methods

To understand what draws individuals to the TH lifestyle, a purposive sample of thirty individuals within three different age categories (18–34, 35–54, and 55 and older) was created. Only individuals who already live or were very interested in living in a tiny house as their primary place of residence were included in the sample. Participants were recruited from an email list provided by Eagle Ridge

---

3　Rodriguez (2017, p. 1) views minimalism as "a broad array of practices that have been labeled differently at different historical moments". By broadening the term, however, he inflates much of the work on the "voluntary simplicity movement" that documents its unique character. Cherrier and Murray (2002) contend that minimalism and voluntary simplicity are distinct movements that may share some of the same grievances but offer two very different anti-consumerist pathways.

Building LLC (a regional tiny house builder in the South) and several social media groups on Facebook and Reddit (i.e., Tiny House People, Tiny House Life, DFW Tiny House Enthusiasts).

During the summer of 2017, 30 semistructured phone interviews were conducted, touching on a wide range of TH issues (e.g., meaning of a tiny house, obstacles to living in a tiny house, desire for community), but primarily focused on reasons and motivations for people wanting to live "tiny". A range of different demographic indicators were also collected[4]. Each participant was asked for consent prior to being interviewed and were given pseudonyms to protect their identities. All interviews (average length: 69 min) were then recorded and transcribed in NVivo 12 for further data analysis. Using an inductive–deductive content analysis ([Bryman 2016](#)), all interview transcripts were first coded using an open coding technique leading to sixteen themes. The themes were further condensed via an axial coding approach leading to a six-theme structure. During the last stage of the coding process—selective coding—the connections among the themes were explored more deeply, which culminated in early contours of the theoretical framework for tiny house living. The coding process produced 23 sources for *Prior Strains*, 28 sources for *Financial Security*, 25 sources for *Autonomy*, 18 sources for *Meaningful Relationships*, 30 sources for *Simple Living*, and 25 sources for *New Experiences*.

Reasons for simple living were further classified as personal motives and mixed motives. To more carefully distinguish among different motivators for simple living, we coded participants' responses into two different motivator themes: Personal reasons (e.g., to save money) and environmental reasons (e.g., to reduce their footprint). Once this coding was completed, we then assigned each interviewee to one of the following three categories: *Simple living category 1 (personal reasons only)*, *simple living category 2 (mixed motives I: Primarily personal reasons)*, and *simple living category 3 (mixed motives II: Primarily environmental reasons)*[5]. Interviewees that only described how going tiny would benefit them personally were assigned to *simple living category 1*. Interviewees who made references to environmental reasons (e.g., use solar power) but did so for primarily practical or utilitarian purposes (e.g., to save money and live off-grid) were assigned to *simple living category 2*. The remaining people who stressed green considerations (e.g., reduce consumption, use alternative energies) for primarily collective and other-centered benefits (e.g., "to save the planet" or reduce "one's footprint" for future generations) were assigned to *simple living category 3*. The final coding scheme was then used to develop the theoretical framework for tiny living (see next two sections).

## 4. Results

While the individual appeal of the tiny house lifestyle can be traced to a multifaceted, often nonlinear, and highly complex reevaluation process, one that mirrors the principle of 'edit' in minimalism ([Babauta 2009](#)), the storyline that emerges from the interviews is much more intriguing. The narrative accounts of the interviewees suggested that many of the individuals did not actively seek the lifestyle but merely stumbled upon it. In a culture that stresses success, hard work, and material comfort while at the same time encouraging individuals to strive for happiness and fulfilment ([Williams 1970](#); [McMahon 2006](#)), the TH lifestyle seems to offer an experiential lifeboat to navigate the ever-changing waters of our time *and* a "new" moral compass to orient their lives. TH journeys often began when individuals experienced struggles that forced them to reexamine their lives (prior strains).

---

[4]   Information on the interviewee's race, religious affiliation, education level, household income, and political orientation was collected. Rather than using a premade survey item for political orientation, we allowed the participants to self-identify (and coded their responses into a broad conservative, liberal, libertarian, and "other" category). We thought that this information could prove useful in providing further insights as to whether "ideological leanings" create particular affinities to tiny living. This relationship proved to be difficult to disentangle empirically.

[5]   To establish the primacy of choice for each individual (and hence assign interviewees to one of the three categories), we considered three key indicators: (1) How early in the interview someone talks about environmental reasons (e.g., first response to the question or later in the interview), (2) the amount of time someone spent talking about personal as opposed to environmental considerations, and (3) the primary reason given for a particular choice (e.g., personal benefits versus benefits for the environment, community, society or others).

In the search of a real "way out" or a "better life", as the following discussion will show, interviewees came to see financial security, personal autonomy, meaningful relationships, simple living, and new experiences as their new milestones toward happiness (see Figure 1).

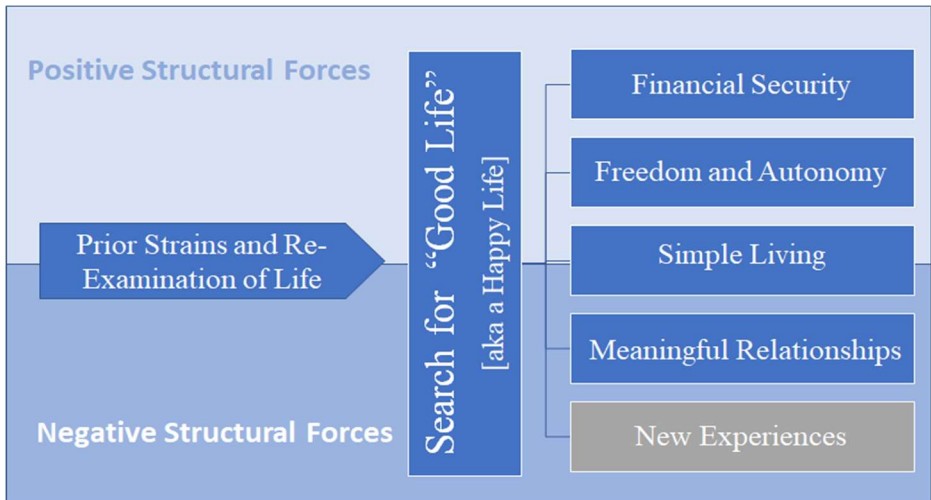

**Figure 1.** Simple conceptual model for the tiny house (TH) lifestyle appeal.

*4.1. Demographic Characteristics*

The sample of TH enthusiasts (*n* = 30) was composed of 53% female and 47% male. Participants came from 12 different US states; most lived in the Southern United States. The racial breakdown of the sample was 83% White, 3% Asian, 3% African-American, and 10% other. Eighty percent of the participants reported annual household incomes of less than $75,000, 30% between $50,000–$74,999, and only 20% making more than $75,000. Political self-identification of the sample was as follows: 37% liberal, 17% conservative, with the remainder claiming other. Answers within the category of "other" were too diverse to group into any one cohesive category. Most participants (76%) reported having completed vocational training or some sort of post-secondary education. While all participants in the study were committed to the tiny house lifestyle, they were at various stages in their TH journeys. In addition to people already living in a tiny house (33%), the sample also includes individuals that were only in the planning and building stages (also see Table 1).

**Table 1.** Select demographics of TH enthusiasts.

| Pseudonym | Age | Gender | Race | Political Orientation | Household Income | Education | Religion | TH Status |
|---|---|---|---|---|---|---|---|---|
| Artemis | 18–34 | female | white | conservative | $50,000—-$74,999 | college degree | Christian | building stage |
| Frank | 18–34 | male | white | liberal | less than $25,000 | college degree | Agnostic | living in TH |
| Namor | 18–34 | male | white | other | $50,000–$74,999 | post-graduate degree | Christian | living in TH |
| Barbara | 18–34 | female | white | other | $25,000–$34,999 | college degree | Atheist | planning stage |
| Tessa | 18–34 | female | other | other | $25,000–$34,999 | college degree | Agnostic | planning stage |
| Ashley | 18–34 | female | white | conservative | $25,000–$34,999 | college degree | Christian | planning stage |
| Rick | 18–34 | male | white | conservative | $75,000–$99,999 | some post-graduate work | Christian | planning stage |
| Luna | 18–34 | female | white | liberal | $75,000–$99,999 | college degree | Atheist | planning stage |
| Natalie | 18–34 | female | white | liberal | $25,000–$34,999 | some college | Atheist | planning stage |
| Bernie | 18–34 | male | white | libertarian | $100,000–$149,999 | some college | Other | planning stage |
| Tom | 35–54 | male | white | other | $75,000–$99,999 | post-graduate degree | Spiritual | living in TH |
| Ben | 35–54 | male | white | liberal | $50,000–$74,999 | post-graduate degree | Agnostic | living in TH |
| Larry | 35–54 | male | other | other | $50,000–$74,999 | less than high school | Spiritual | building stage |
| Greta | 35–54 | female | Asian | liberal | $50,000–$74,999 | college degree | Agnostic | planning stage |
| Jenny | 35–54 | female | white | libertarian | $35,000–$49,999 | trade/technical/vocational training | Christian | planning stage |
| Nancy | 35–54 | female | white | other | $50,000–$74,999 | college degree | Christian | living in TH |
| Sebastian | 35–54 | male | white | liberal | $25,000–$34,999 | post-graduate degree | Other | living in TH |
| Dan | 35–54 | male | white | conservative | $35,000–$49,999 | some post-graduate work | Other | building stage |
| Brittany | 35–54 | female | black | liberal | $50,000–$74,999 | post-graduate degree | Christian | planning stage |
| Jim | 35–54 | male | white | liberal | $50,000–$74,999 | high school | Spiritual | living in TH |
| Jane | 55+ | female | white | other | $25,000–$34,999 | college degree | Spiritual | planning stage |
| Mary | 55+ | female | white | liberal | $75,000–$99,999 | college degree | Christian | planning stage |
| John | 55+ | male | white | other | $25,000–$34,999 | post-graduate degree | Spiritual | planning stage |
| Tim | 55+ | male | white | liberal | $50,000–$74,999 | post-graduate degree | Spiritual | living in TH |
| Canan | 55+ | male | white | liberal | $75,000–$99,999 | some college | Other | living in TH |
| Venus | 55+ | female | white | other | less than $25,000 | post-graduate degree | Buddhist | planning stage |
| Morgan | 55+ | female | other | other | less than $25,000 | some post-graduate work | Spiritual | living in TH |
| Samantha | 55+ | female | white | other | $35,000–$49,999 | some college | Christian | building stage |
| Shirley | 55+ | female | white | conservative | $35,000–$49,999 | post-graduate degree | Christian | planning stage |
| Abraham | 55+ | male | white | other | $35,000–$49,999 | high school | Christian | planning stage |

*4.2. Prior Strains and the Need for a Reexamination of Life*

For 23 of the interviewees, the journey into the tiny house lifestyle began with the turmoil of an existential crisis and/or the challenges brought on by major life events. These strains pushed individuals to reflect harder upon their lives and drove them to seek out alternatives to their current living circumstances—a search that eventually led them to "discover" the TH lifestyle.

*Existential crises.* In Ashley's case, for example, a general "abundance of items" she had collected over years that were "weighing . . . [her] down" triggered her process of introspection, which led her to realize that more "stuff" does not equal happiness. For others, like Natalie and Jim, it all began when they finally realized that the bigger-is-better philosophy everyone told them to embrace turned out to be the root of their unhappiness. Natalie's combined lack of happiness as well as the unhappiness she witnessed within her family fueled her towards making a change in her life. Meanwhile, Jim struggled to grasp the issue of excess unused space whilst attempting to maintain a healthy work–life balance.

> I can't understand why I've spent so much money this long and why my family is doing the same thing . . . all under the illusion that something is being gained, but really, they just have a bunch of junk in their basement that they never look at and they're all really unhappy with their houses. –Natalie (age 18–34)

> In the wintertime we were paying to heat rooms . . . we never used . . . in the summertime, we were paying to keep them cool . . . sometimes I didn't even look into . . . [the] rooms for . . . months . . . because it was basically storage . . . [I] kind of felt like this . . . [for] ten years and maybe back then it was like . . . a mid-life crisis . . . I was working a job working ten twelve, thirteen, fourteen hours a day . . . trying to maintain a home and vehicles and . . . never having any time . . . to enjoy . . . what I wanted to do . . . you get up in the morning, go to work, come home at night, eat dinner, take a shower, go to bed and . . . you know . . . that's just how it was. –Jim (age 35–54)

*Major threats as triggers.* Many interviewees like Ashley, Tessa, and Artemis discussed major financial and economic stressors as important reasons for paying closer attention to the tiny house lifestyle.

> Losing my house played a role. The economy collaps[ing] played a role. The fact that I have so much education and I struggle to find a job, all those things back in that time played a role in my interests in living smaller. –Ashley (age 18–34)

> [H]aving my own property and not . . . [being] shackled down for my whole life . . . I feel like that's . . . what really drew me to . . . seriously pursue this [tiny house living] . . . I have a serious amount of student loans and having that debt follow me around wherever I go . . . is very crippling . . . I can't even imagine bringing on extra debt with a mortgage. My credit score's ridiculously low and that's why I'm having to save money. –Tessa (age 18–34)

> [R]ight now we're at a standstill of we can make it work . . . with our 15-year mortgage . . . money is super tight . . . we can re-finance into a 30 year, and that frees up a bunch of money, [but] . . . me being the nerd, I don't want to do that. –Artemis (age 18–34)

While debt and financial difficulties characterized many of the narratives, some, like that of Abraham, talked more about how major life events, such as divorce and children moving out, pushed them to seek out alternatives.

> [Divorce] it's probably one of the key factors . . . [the] gal . . . was the love of my life . . . she asked for a divorce and . . . that was kind of a shock, I kind of dusted myself off and . . . said okay I really got to make a change here. So, minimalizing my lifestyle was the key ... So that's where the whole tiny house . . . really came into play. –Abraham (age 55+)

The most common struggles faced by TH enthusiasts suggest that their interest in the TH lifestyle may have been driven—at least initially—more by necessity than by choice. Many interviewees stressed that to them, tiny living seemed to offer a real "way out" of situations in which they felt "stuck" or "trapped". As much of the early work in the sociology of collective behavior has demonstrated, strains are only a necessary but often not a sufficient condition for people to act (Smelser 2011). This is true for collective as much as it is for individual actions. In situations where negative life experiences trigger motivational eddies and induce individuals to make decisions (Fogg 2009), the belief in the efficacy of the action became paramount. Social circumstances further seemed to mediate these deliberative processes, suggesting that for TH enthusiasts immersed in more supportive environments, the transition into the TH lifestyle came easier (Fishbein and Ajzen 1975; Ajzen 2011). Through a process of ongoing reflection, research, and exploration of this "fringe" lifestyle, individuals in this study slowly came to realize that embracing tiny houses not only meant a solution to their problems, but a path toward what many were longing for: A chance at living a version of the *Good Life*.

*4.3. Core Motivators and the Search for the "Good Life"*

When initially asked to explain their reasons for pursuing a tiny house dream, participants often expressed interest in "downsizing, getting smaller", or simply sought a "cheap[er]", more "novel and cute" house. While the longing for simpler, more self-reliant, and more affordable lives was present in the people's "conversion narratives", the true appeal of the lifestyle seems to come from a deep-seated, at times unarticulated, search for the *Good Life*. While some followed a Eudemonic road that stressed the collective good (Rowe and Broadie 2002; Skidelsky and Skidelsky 2012), many of the other interviewees described their search in more personal terms, such as wanting to "live well" and/or "be happy". In the minds of the interviewees, the TH lifestyle offers a blueprint on how to: (1) Regain ontological security (i.e., financial security), (2) retake control over their lives (freedom and autonomy), (3) nurture deeper relationships with friends and family (meaningful relationships), (4) embrace a simpler life (simple living), and ultimately (5) pursue activities that give them pleasure, satisfaction, and happiness (new experiences).

4.3.1. Motivator 1—Financial Security

Deeply affected by the strains in their lives and having found new hope in the financial mantra of the tiny house movement, all but one participant made direct or indirect references to wanting to be financially secure. Participants yearned to "reduce expenses", "save money" or "get out of debt" (also see Figure 2). Some hoped to relegate financial liabilities, such as "student loans", "car payments" or things that make it "hard . . . to get started with . . . life", to a thing of the past. Debt and constant financial obligations are the root of the problem, as Rick explains.

> [W]hen you look at the way that a lot of people live their lives . . . they're tied to . . . student loans and . . . mortgages and all of this kind of stuff and . . . you work to have something to impress somebody you don't like, for what purpose? –Rick (age 18–34)

Feeling trapped and seeking exit strategies, TH enthusiasts followed one of two interconnected strategies to gain financial security and, ultimately, a "debt free lifestyle" by reducing financial liabilities and/or cutting down expenses.

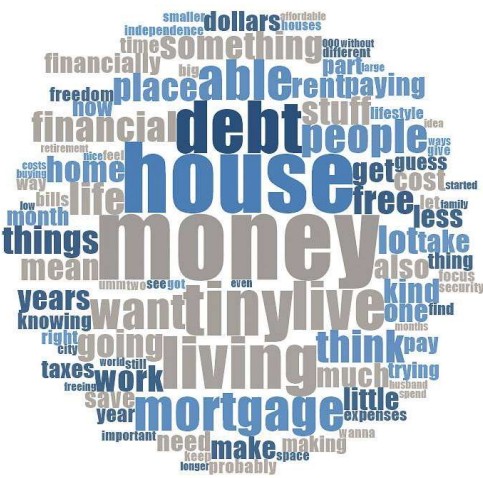

**Figure 2.** Word cloud for Motivator 1.

*Reducing financial liabilities.* One strategy mentioned by interviewees to achieve financial security was eliminating financial obligations, such as mortgages and unnecessary expenses, to "rapidly get out debt". While many TH enthusiasts already valued frugality, some, as Ben poignantly put it, had to undergo a "lot of deprogramming" to change their attitudes toward finances. For older interviewees, concerns about retirement provided an added incentive to move toward an "essentially debt free" lifestyle. Eliminating debt also meant reframing mortgage and rent into major stumbling blocks toward financial security and the Good Life. Jane, John, and Larry put this best when they say:

> I don't want to pay the rent . . . I would rather own my own than pay . . . rent so . . . the bottom line for me is . . . [I want to] live a more debt free [and] responsible life. –Jane (age 55+)

> I think I would be reducing, certainly I would not have a mortgage I wouldn't . . . rent or if I did, maybe [at a] much reduced [cost]. –John (age 55+)

> [A] 30-year mortgage ties a person down . . . one is pretty much enslaved to one's home . . . at 51 years of age, I don't want to tie myself to a 30-year mortgage. –Larry (age 18–34)

*Reducing expenses.* Efforts to eliminate existing financial liabilities often went hand in hand with attempts to drastically "reduce expenses". To regain control over their finances, as well as have more "money to play with", many TH enthusiasts, as Dan pointed out, had to learn or relearn "how to better regulate and budget . . . [their] money so [they] don't need as much." Others began "go[ing] to the library . . . to keep . . . costs down at the house", like Natalie. To avoid living in a tiny house community because they "charge anywhere from $300–400 a month for [a] plot", like Samantha, many TH enthusiasts decided to go back to the "basics". Going tiny also meant a drastic reduction of housing costs. Ben, for example, managed to go from paying "$1000 dollars [for] rent to $300 dollars [per month]"—a game changer for him. Others, like Tim, witnessed a fundamental shift in their budget and disposable income. He describes both a liberating and empowering experience as follows:

> My actual expenses yearly are a little less . . . than $15,000 so there's a $35,000 savings [from my yearly income] that I'm just putting in the bank and it's making money for me so I can do things . . . travel oversees . . . Give gifts to my family you know. It's just a freeing experience. –Tim (age 55+)

Whichever strategies interviewees used, they were all—to some extent—drawn into the tiny house lifestyle by a promise of financial security. While less central in the philosophical understanding of its sister movements, such as voluntary simplicity, financial freedom can be seen as a precondition to ontological grounding (Grigsby 2004; Chhetri et al. 2009; Millburn and Nicodemus 2015). The wish to feel "secure" often came packaged with a deep desire to regain more freedom and autonomy in their lives.

### 4.3.2. Motivator 2—Freedom and Autonomy

Eighty three percent of participants expressed a deep desire to take back their lives. Many TH enthusiasts saw living tiny as a gateway to "being in control of . . . life" or at least help to "redefine where the quality of . . . life [should be]." While often couching it in the language of wanting "more time and money", many interviewees expressed "want[ing] out of it". Seeing their mortgage, debt, long work schedules, and the consumerist lifestyle as contemporary forms of "slavery", they longed to break *free from* oppressive structural forces that they felt held them back from the rich and fulfilling lives they longed to have (also see Figure 3). To attain this true sense of autonomy and overcome the grips of what Brittany called the "American lie", many TH enthusiasts stressed that one needed to go beyond the simple idea of *freedom from* and embrace the much richer and much more genuine *freedom to* orientation to life (Fromm 1941).

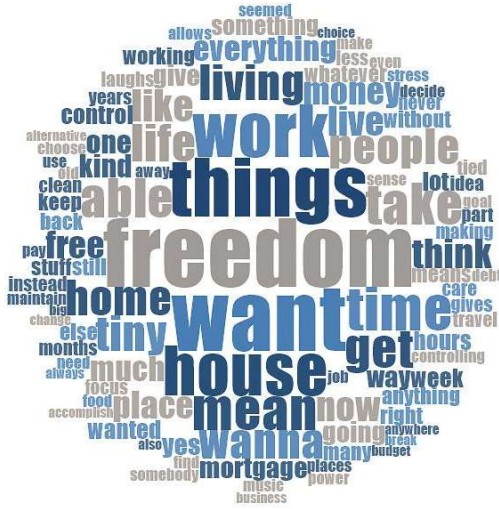

**Figure 3.** Word cloud for Motivator 2.

"*Freedom from" narratives.* Throughout the interviews, it was evident that regaining freedom from the strains of contemporary life is a central factor in the overall appeal of the tiny house lifestyle. Natalie, for example, spoke of how no longer having to pay rent not only made her feel more secure, but how living tiny gave her a new "freedom from him [her landlord]" that was "huge". While Rick echoed Natalie's sentiments of "never want[ing] to be trapped in a house [rent/mortgage]", others discuss how "chasing after the dollar bill" and being stuck at work all week—the cultural prerogative to success so deeply embedded in American culture—were in fact hindrances to making their own decisions of how to live. Not feeling weighed down by a "lot of things", having the "freedom from obligations" allows people to "have time to [themselves]." Sebastian, Tom, and Dan provide further insights into how going tiny liberated many from the daily *ought to* and the self-permissive *want to*:

> Well, as opposed to continuing within a job I hate, making a lot of money . . . I got to a point where, you know I'm working sixty hours a week, I'm making a good salary, but I don't really love it . . . [today] I'm no longer a slave to my job. So, I have you know twenty hours or more extra in the week to live or do things that I want to do. Instead of being stuck in Jobville forever. –Sebastian (age 35–54)

> I don't need to work 40 and 50 and 60 h a week and be able to still save and invest for you know for my future, but my monthly costs are so minimal . . . So yeah so some of it is freedom, but definitely economic and being part of you know being impacted by the economic downturn. –Tom (age 35–54)

If I 'm not drowning in debt or having a whole bunch of stuff to maintain then I'm free to just live, that's what's attractive about that [the TH lifestyle]. I'm free to go do things, I'm free to get out there in life . . . and do things that I want to do. –Dan (age 35–54)

*"Freedom to" narratives.* Simply breaking free from the cultural and economic shackles, however, was not enough for many of the participants. Most interviewees wanted to regain their full freedom and autonomy to "enjoy life more". Tim, for example, argued that the main driving factor in his decision to go tiny was "freedom, freedom from everything and [being] free to choose what to do, when to do it, where to do it and with whom." Bernie echoes these views by pointing out that what the TH lifestyle "really boils down to [is that] . . . you can take your resources and do the . . . things that you want to do." Put another way, what many of the interviewees sought was the unencumbered right and ability to take back control over important decisions which affected their lives. For some, like Brittany, it meant being able to choose how to structure her day, how "much [or] as little" to work and what clients to take on. For others, like Jim, it came down to not having to "worry about . . . get[ting] home from working late" and going for "a [bike] ride or go[ing] away for a weekend". He feels that now, he could simply decide to go without having to "worry about making arrangements [to] take time off" or being accountable to anyone other than himself.

Feeling captive of a cultural DNA that maps ideas such as individualism, freedom, material comfort, hard work, achievement, success, and conformity to the cultural norms onto our psyche (Williams 1970), many participants saw living tiny as an act of self-emancipation and an opportunity to (re-)create new experiential spaces to be the true masters of their destinies. While striving for ontological security helps TH enthusiasts to alleviate their financial worries, it is the endeavor for autonomy—or "freedom"—that gives them the feeling they can engage in behaviors that are "self-endorsed and congruent with their [own] values and interests" (Weinstein et al. 2012).

### 4.3.3. Motivator 3—Meaningful Relationships

Many TH enthusiasts also yearned to interact with people and build relationships (also see Figure 4). Many alluded to the fact they had "forgotten . . . how to talk to people" and "lost . . . [the] connection to other people." Larry, for example, laments how the "family unit . . . has been disrupted" because nowadays "there is not a lot of family time." In his eyes, people do nothing more than "work [to] pay taxes, . . . utility bills, . . . insurance bills [and] . . . mortgage." Freeing them from constraints placed upon them by contemporary life, the TH lifestyle is seen as opening up new horizons "to meet people" and spend "more time with each other". To some, living tiny even promotes communication, because space makes people "work out the problems right then and there."

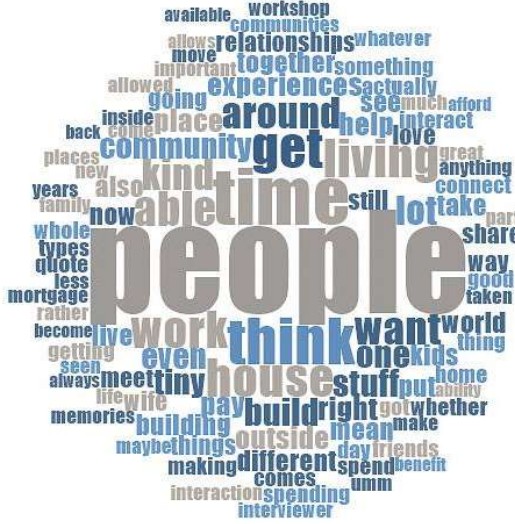

**Figure 4.** World cloud for Motivator 3.

TH enthusiasts see their lifestyle as a tool to allow new relationships to form "naturally", while at the same time strengthening existing relationships. Especially for interviewees struggling with their relationships, the appeal of the going tiny was often a "game changer". In Nancy's case, for example, living tiny gave her the chance "to move . . . [back home] and caretake [*sic*]" her parents "in the last years of their lives"—something she feels she would not have been able to do otherwise. Like Abraham, Samantha and Barbara express their desire for more meaningful relationships in the following two passages:

> I'd personally would love to get to know 'em [other people], I'm a people person, I love talking to people and I love to hear peoples' experiences. –Samantha (age 55+)

> [S]o the experiences with each other and the growth that we'll be able to experience as a couple hopefully because we're so focused on each other and these experiences I think it's going to be something that . . . spreads to other people around us. –Barbara (18–34)

Interest in mending and creating new relationships potentially signals a course correction toward a happier, more fulfilling life. Whether this focus on relationships is merely an extension of the wish to have more experiences remains to be seen. What is clear, however, is that becoming a TH enthusiast is not merely a story of personal transformation, but one intricately intertwined within the social circumstances in which the individuals find themselves.

### 4.3.4. Motivator 4—Simple Living

All interviewees stressed that they wanted to live a "simpler lifestyle in a smaller space". For many, this meant finding ways to reduce their belonging to the "most essential", letting go of "nick knacks", and/or reconsidering the "stuff" they bought or were "bringing . . . home". A closer analysis of the interviews, however, suggests a more nuanced picture as to why people want to live simple. While all interviewees gave reasons for how simplifying their lives and going tiny benefited them personally, some stressed these personal motives exclusively, while others discussed mixed personal and environmental motives (also see Figure 5).

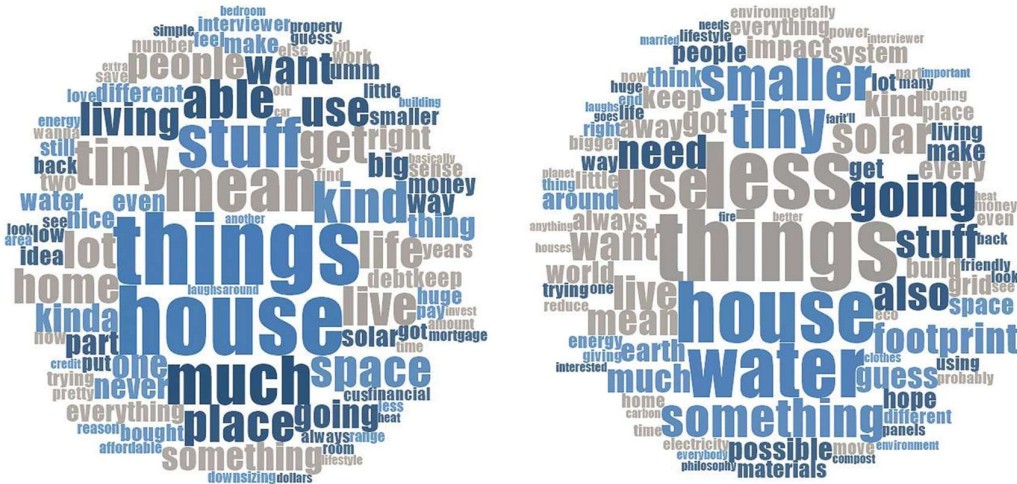

**Figure 5.** Word clouds for Motivator 4. The left cloud represents most frequent responses in the personal motives' category. The right cloud captures word frequencies for the mixed motive category. Note: The word "things" in both clouds occurs in interview contexts where people talk about wanting to "get rid of", "reduce" or "do" things with their newly found freedom.

*Personal Motives.* More than half of the interviewees (17 out of 30) gave personal reasons for wanting to live more simply. These TH enthusiasts wanted to "get rid of stuff . . . they never use[d] and . . . need[ed]" in order to have more functional homes, simplify their daily lives, and save money

in the process. Many of them expressed the desire to create homes that "worked" for them. Others couched their reasons for going tiny in a deep-seated desire to lead less "stressful", less "complicated", and more "efficient" lives. Frank, for example, stressed that "with downsizing comes efficiency", while Ben and Luna commented how letting go of "old clothes, old boxes of stuff" made their lives "more simple". The focus on the practical benefits of the TH lifestyle was often intertwined with a financial rationale. Morgan, for example, emphasized that she could have gotten "electricity where [she] lives" but decided to buy "solar lights" because she "didn't want to pay the bill" on her low income. Bernie viewed "solar" as a method to "offset [his] energy bill" and "make . . . money from the power company". Commenting "I'm not some environmentalist by any means trust me I'm a hundred percent capitalist", he stressed the benefits of living "debt free . . . quicker" and achieving a "potentially off-the-grid" lifestyle. Brittany also stated that saving money and downsizing were her reasons for going tiny and that her desire to "use solar . . . and take advantage of wind energy" stemmed from wanting to be more self-reliant. Dan echoed the financial views by saying that going tiny will "save [him] a lot of money". When other TH enthusiasts in this category touched on green aspects, they stressed functional, practical, and financial aspects that they saw as benefiting them as individuals.

*Mixed Motives.* Less than half of interviewees (13 out of 30) expressed mixed motives for wanting to go tiny. Of those thirteen, six commented that their desire towards simple living comes from wanting to be "environmentally aware and conscious". They stressed the desire to use renewable energies and live a greener lifestyle to "tread lighter on this earth" or "be true to the earth". John, for example, wanted "the best technology possible to reduce [his] carbon footprint and live in harmony with nature". Larry considered the ability to live greener his "biggest push", while others, like Jenny, found inspiration in their faith by reminding us "we should [all] be good stewards to . . . God's creation." For Barbara, personal and environmental reasons came together in what she dubbed a "double win".

> [I]f we can . . . live off renewable energies then essentially our bills should be nothing . . . so we're hoping that we can do the earth a favor and it can do us a favor by lowering our bills. –Barbara (age 18–34)

The remaining seven TH enthusiasts made references to both personal benefits and environmental considerations. When analyzing the interviews more carefully, however, it became clear that environmental reasons were secondary. Participants in this group had a general understanding of sustainability, but the nature of their answers illustrated that they were "not super green". Furthermore, Tim, Ashley, and Abraham touched on the practical and collective benefits of green technologies, but considerable time elaborating how "downsizing" and becoming "a minimalist" "works" and "ma[de] sense" for them. Taken together, the interviews suggest most TH enthusiasts turn to the lifestyle for reasons that are primarily or exclusively personal. TH enthusiasts with predominately environmental motivators are rare.

### 4.3.5. Motivator 5—New Experiences

Finally, all TH enthusiasts expressed a passionate desire to have more experiences within the "simplicity" of their newly found lifestyles. With the time, resources, and license to "enjoy other things in life" or live "life to the fullest", people wanted to "travel and see a lot of things", pursue hobbies or passions as well as simply try out new things (also see Figure 6). TH enthusiasts sought to "spend more time outside", "do things", and reconnect with their passions like "studying", "biking", "hang gliding", "sky diving", "rock climbing", "whitewater rafting" or "attend[ing] . . . music festivals". Others wanted to "see all the best Broadway show[s]" and hear the "best music" or eat the "best food". Most interviewees stressed that "happiness comes from experiencing things" and that life "is all about the experience", and emphasized that tiny living allows them, often for the first time, to be "in the moment" and do the things they truly desired to do. Those earlier in their journeys often remarked that others on the "path" "just seem[ed] like they were happier" because "they

didn't have things to be stressed over." Quotes from Nancy, Barbara and Bernie nicely illustrate this trading-experiences-for-things rationale.

> [I]t's all about experiences and . . . the time that you spend with people or the time that you spend having those experiences versus the time that you are taking to manage your possessions. –Nancy (age 35–54)

> [T]here are so many people that have a focus on the things that bring you happiness and I want my experiences to bring me happiness. –Barbara (age 18–34)

> I'm an experience person. I have nice things, but I don't love my things . . . I'd rather spend $2000 on a trip to South America than on a TV . . . I have a $300 TV . . . I'd rather take that other $1500 and spend it on an experience. –Bernie (18–34)

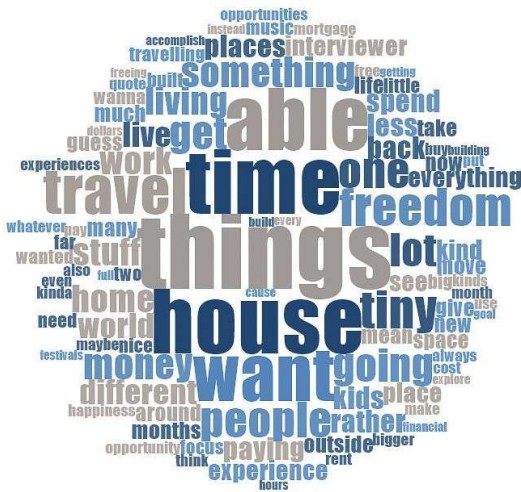

**Figure 6.** Word cloud for Motivator 5.

While TH enthusiasts strive for more fulfilling and richer experiences as ways to attain happiness, over half of the interviews talked about happiness in terms of "feeling good" and less so as "being good" (McMahon 2006). This—as will be argued in a later section—may reflect the more individualistic zeitgeist of our times.

## 5. Discussion

While replicating insights from earlier studies (Mutter 2013; Boeckermann et al. 2017) that situate the motivations for tiny living in simplicity, sustainability and environmentalism, cost, freedom and mobility, sense of community, and interest in design, the paper offers a new conceptual framework for understanding the wider appeal of the TH lifestyle. It argues that the TH movement—and the lifestyle it promotes—constitutes a new attempt in answering an age-old question: How does one live a *Good Life*? Riding out the waves of the 2008 economic recession and struggling to come to terms with the complexities of the new millennium, individuals embrace a TH lifestyle because they see it as a vehicle toward the *Good Life*. While some TH enthusiasts want to live environmentally conscious lives, issues of sustainability are often secondary. What seems to make the TH lifestyle unique in the eyes of its followers (and, as such, sets it apart from its sister movements) is its more individualistic and pragmatic view of simple living, its focus on autonomy, as well as its emphasis on experiences and relationships as a means toward happiness.

### 5.1. Simple Living as Functional Pragmatism

Like voluntary simplifiers and minimalists, TH enthusiasts embrace a version of simple living. They share their sister movements' distaste for the widespread consumerist lifestyle (or their rejection of

the consumption of goods) and seek alternatives to the conspicuous consumption many grew up with (Etzioni 1998; Cherrier and Murray 2002; Grigsby 2004). While many voluntary simplifiers strive to live up to the philosophical core of their tradition (Etzioni 1998; Ross 2015), for TH enthusiasts, being green is one of the many choices they can—and sometimes do—make. Simple living, to them, involves small spaces that are functionally adequate and experientially liberating. Echoing the rugged utilitarianism of contemporary minimalists, TH enthusiasts embrace downsizing, decluttering, and downshifting because they tend to equate the Good with the Easy and the Practical. While minimalists view downsizing as an ongoing process (Babauta 2009; Millburn and Nicodemus 2015), TH enthusiasts may buy new items that they think will enhance their experiences (e.g., buying new sports gear to have fun). In fact, if TH enthusiasts do have to make choices, the pragmatism of simple living and the primacy of experiences tend to trump the essentialism of minimalists and the value-adherence of voluntary simplifiers.

Sustainability and ecological consciousness may make up core orienting values for practitioners of voluntary simplicity (Elgin and Mitchell 1977; McDonald et al. 2006), but environmental considerations are often less salient in TH enthusiasts' narratives. Using alternative energies (like wind and solar), installing composting toilets, or engaging in practices that reduce participants' carbon footprints, for many, is as much about pursuing a more self-reliant off-the-grid type of lifestyle, simplifying daily life, and/or saving money as it is about protecting the environment. This may be, in part, because TH enthusiasts' views on simple living are deeply influenced by contemporary minimalist ideas—ideas that tend to lack a systematic focus on environmental concerns. Instead, minimalist manifestos shift attention to the attractiveness of personal choice and the liberating effects that living with less can have for the individual (Babauta 2009; Millburn and Nicodemus 2015). While the focus on the "practical" and "functional" appears to be central in the TH appeal, living simply also resonates with individuals because it seems to provide them with a means toward financial independence and, ultimately, a sense of ontological security so paramount in the conceptions of the *Good Life* (Skidelsky and Skidelsky 2012). Living simply, however, not only grounds the individual and gives them back a sense of agency. It also frees up time and resources that TH enthusiasts then direct toward having more experiences and nurturing deeper relationships.

## 5.2. Strive for Autonomy

TH enthusiasts view "being in complete control" of their own choices as paramount to the *Good Life*. The freedom to choose—so deeply entrenched in the American experience (Thoreau 1910)—is a value that is also shared by followers of VS and minimalism (Elgin and Mitchell 1977; Leonard-Barton 1981; Craig-Lees and Hill 2002; McDonald et al. 2006; Millburn and Nicodemus 2015). While all three lifestyles emphasize the importance of autonomy, self-determination, and personal freedom, they come at these ideas from very different directions. Unlike TH enthusiasts, many voluntary simplifiers and minimalists begin their journeys into simpler living with a conscious act of rebellion. Often coming from more privileged backgrounds, they exercise their structural autonomy to rebel against the perceived emptiness of consumer culture, not because they have to, but because they can (McDonald et al. 2006; Gardner 2015).

While simple living may eventually become the existential modus operandi, for many TH enthusiasts, the decision to go tiny is often much less voluntary. Often struggling to regain financial independence, many TH enthusiasts downsize in order to save money and/or free up resources. Like minimalists, TH enthusiasts want to be control over their lives in order to "increase personal life satisfaction" and explore "new opportunities" (Rodriguez 2017, p. 8). While proponents of all three lifestyles reject consumerism, voluntary simplifiers spend much more time trying to address the environmental impact of their choices or attempt to challenge the underlying structural and cultural forces (Grigsby 2004). Tying autonomy to self-direction, Elgin and Mitchell (1977), for example, discusses the perceived need by voluntary simplifiers to relocalize aspects of their lives (e.g., food production) in order to break free from the corporate control and the system that they feel enslaves

them. Asking some of the big picture questions that many minimalists and TH enthusiasts tend to eschew, they are more likely to spend their time volunteering, participating in community building, or gearing their efforts toward social or political change (Etzioni 1998).

Environmental reasons, such as "saving the planet", can and do play a role for some TH enthusiasts. This environmental orientation, however, does not seem to motivate most of the TH enthusiasts. What they all do share, however, is a general appreciation for the functional, pragmatic, and liberating aspects that their choices to live tiny entail. Drawn to the much more individualistic philosophy of contemporary minimalism, which seems to sit at the center of the TH universe (Millburn and Nicodemus 2015), TH enthusiasts seek the autonomy that the lifestyle promises. Whether this freedom gets directed toward personal or collective ends is, thus, secondary. Control over one's choices seem to matter more than the nature of that choice. While the strife for simpler living and the longing for autonomy provides an ontological grounding for the *Good Life*, it is the pursuit of meaningful experiences and relationships that gives it its existential substance.

*5.3. Centrality of Experiences and Relationships*

Rich experiences and deep meaningful relationships are the center of the TH psychosocial cosmos. Unlike voluntary simplifiers who—in addition to their environmental interests—place much more importance on spiritual development, inner growth, and intellectual pursuits (Elgin and Mitchell 1977; Grigsby 2004), TH enthusiasts identify the "Good Life" primarily with new and more meaningful experiences. Having lived "lives of quiet desperation", to quote Thoreau (1910, p. 8), many of them try to use their newly found autonomy to live in the Now. Like adherents of voluntary simplicity and minimalism (Grigsby 2004; Babauta 2009), they decouple happiness from the lure of materialistic possessions and channel their resources into experiences that offer them a deep sense of satisfaction, contentment, and purpose in life. Many proponents of the TH lifestyle also intuitively know that the *Good Life*—the happy life—is impossible without the ability to nurture deep meaningful relationships, although the desire to connect and reconnect with others often remains subsumed in a much deeper experiential quest for a complete reorganization of everyday life. To invite further discussion and research, we argue that the centrality of experiences for TH enthusiasts is not only a cultural byproduct of our times, but also the psychosocial consequence of the nature of experiences themselves.

While showing several philosophical continuities with earlier lifestyle movements (Etzioni 1998; Rodriguez 2017), the "sudden" appeal of the tiny house lifestyle and its love affair with experiences may be seen—at least in part—as a byproduct of the much larger societal transformations within post-industrial societies over the past few decades. Gerhard Schulze, a German sociologist, is often credited with having developed the first systematic treatise on how Western economies and cultures have begun to shift from a consumption of goods toward the commodification, marketing, and consumption of experiences (Pine and Gilmore 1999). In his modern classic *Die Erlebnisgesellschaft: Kultursoziologie der Gegenwart* (*The Experience Society: A Cultural Sociology of the Present*), he argues that the economic growth after World War II in Germany triggered the emergence of a new experience-based society with an "experience rationality", an "experience market", and an "experience milieu" at its core. This new experience society shifted the cultural focus away toward inward-looking lifestyles and planted the simple countercultural message: Enjoy life and worry about money later. (Schulze 2005, p. 63)[6] writes: "the experience society is the project of a beautiful life" in which "experience is the dominant form that defines the meaning of life. . . . [s]lowing down instead of speeding up; less rather than more, uniqueness instead of standardization; . . . doing instead of consuming; acceptance instead of push for gain; . . . more money for travel, less money for TVs, HiFi devices and cars . . . a happiness discourse that has become the new standard."

---

[6]  The authors assume full responsibility for any problems with the quality of the English translation of Schulze's German original.

While Schulze wrote about German cultural trends, other authors have maintained that the cultural evolution in other post-industrial societies have followed similar trajectories (Bryman 2004). Sundbo and Darmer (2008), for example, contend that the push toward happiness and experiential living has led to a massive rethinking of the global capitalist economy. While earlier economies produced goods and services to satisfy customer demand, the "new" economy has begun to sell experiences. Echoing the insights of the Frankfurt School (Adorno 1975), Sundbo (2009, p. 433), goes as far as to argue that the experience industries attempt to "train the public to appreciate . . . experience[s]". These "highly systematized and industrialized" experience packages, Sundbo and Darmer (2008) argue, have even begun to seep into the organizational DNA of public institutions (e.g., museums, municipal cultural centers, and city governments) and voluntary organizations (e.g., sports clubs). Increasingly forced to "operate under market conditions", they have become the new cathedrals of consumption (to borrow and expand Ritzer 2005 use of the term) of the new cultural age in which people are sold packages reminiscent of "an interesting life", "experiences [of] new . . . places", and other forms of entertainment (Sundbo and Darmer 2008, p. 3).

Cast against the broader backdrop of longer working hours, stagnating real wages, and increasing personal debt (Jordan 2017), it may not be not that far-fetched to view the experience-oriented content of the TH universe with its blogs, TV shows, and festivals mirroring the logic of lifestyle packages so carefully crafted by the experience industry. If that were to be the case, it would be somewhat ironic to think that by indulging in these commodified experiences, some of the staunchly anticonsumerist TH enthusiasts may inadvertently become users of these new forms of consumption. The growing commercial interest in tiny houses among builders, developers, and companies selling products (Bahney 2018) may signal either an increasing cultural rejection of McMansions or commodification or the economic capture of the movement and its philosophical affinities for the *Good Life*. Only time will tell.

While the new postindustrial consumer economy may increasingly try to package and sell lifestyle packages, positive experiences with or without a price tag have been shown to lead to personal well-being, promote happiness, and lead to what TH enthusiasts ultimately seek: The *Good Life*. As the large and growing body of the social psychology of experiences, for example, demonstrates, spending money on "life experiences" can produce greater happiness than spending money on new possessions (Van Boven and Gilovich 2003; Howell and Hill 2009; Nicolao et al. 2009). This, however, only holds true in cases where these experiences are evaluated as positive (Nicolao et al. 2009) and shared with others (Caprariello and Reis 2013). The social and relational aspects of the experiences seem to be especially central (Gilovich et al. 2015), as they tend to generate longer-lasting satisfaction (Carter and Gilovich 2012) and are more likely to satisfy deeper-seated psychological needs (Howell and Hill 2009). Since people's memories of the experiences tend to be more positive than the experiences themselves (Mitchell et al. 1997), experiences do powerfully shape people's sense of self (Carter and Gilovich 2012). In recognizing the social and psychological benefits of their new experiences, TH enthusiasts may find an inexhaustible source of motivation toward the simpler, more grounded, and more self-directed TH lifestyle. Ryan Mitchell, an author who writes books about tiny house living, captures the experiential TH essence best when he writes: "When you live intentionally, you realize you have choices—and that those choices empower you to be where you want to be, do what you want to do, and live the life you always wanted" (Mitchell 2014, p. 221).

## 6. Conclusions

While the study offers intriguing insights into what motivates people to adopt this lifestyle, there is much more that remains unknown about the inner workings of the TH universe. Future research should therefore not only try to test the proposed motivational model in a more representative sample, but should also attempt to answer other important questions, such as: What role do social networks (virtual or face-to-face) play in drawing people into the lifestyle and keeping them committed to it? How important are preexisting values and beliefs in getting people to make the "shift"? To

what extent does the "experience economy" shape experiences of TH enthusiasts? How will the TH lifestyle, the nature of living tiny, and the meaning of a "*Good Life*" change as commercial interests and mainstream actors increasingly shape the narratives within the tiny house movement (e.g., commercial builders and developers)? As the theoretical model was developed for the sociocultural context in the United States, future research should not only be conducted within North America, but also in countries that have seen emerging regional TH movements. The persistence, recurrence, and the continued mystique of these countercultural lifestyles calls upon us within the social and behavioral sciences to develop not only a better theoretical understanding, but a more subtle appreciation of how individuals within different historical contexts negotiate their lives.

**Author Contributions:** Both authors have contributed equally to the research and manuscript (i.e.,: conceptualization, methodology, formal analysis, investigation, resources, data curation, writing—original draft preparation, writing—review and editing, visualization, project administration, and securing funding).

**Funding:** This research was partially funded through the 2017 UNG Summer FUSE Program and generous support from the Department of Sociology and Human Services of the University of North Georgia.

**Acknowledgments:** The authors would like to extend gratitude to the participants for their time and willingness to participate. The authors are also deeply indebted to Chelsey Willoughby, Devin Hing, and Danny Hatch for their help with research methodology, data collection and the development of earlier versions of the theoretical model. We would also like thank Codey Collins, Dace Lewis and Valerie Odorico for their role in data collection and help with transcriptions. Finally, we would like to express our gratitude to Samantha King, Alana Berry, Mickey Reese, Daniel Miles and two anonymous reviewers for their helpful comments on earlier drafts of the manuscript. The Institutional Review Board (IRB) of the University of North Georgia approved this study under code 2017-118-COM.

**Conflicts of Interest:** The authors declare no conflict of interest.

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
