# Peer review of "In Search of the “Good Life”: The Appeal of the Tiny House Lifestyle in the USA"

_socsci, doi:10.3390/socsci8010026_

Round 1
Reviewer 1 Report
This is a good and interesting paper on an under-researched topic. I was particularly interested to read about some of the reasons why people choose to live in tiny houses, the importance of 'freedom' as motivator, and thought that the comment on page 13, lines 574 to 579 especially pertinent.
Minor edits are:
p1, l33: suggest add Shearer, H as a reference, although Australian, she is an active researcher in tiny houses (i.e. Towards a Typology of Tiny Houses, H Shearer, P Burton - Housing, Theory and Society, 2018)
p1, footnote; good differentiation
p2, l48, see first bullet point re Shearer (i.e. https://theconversation.com/interest-in-tiny-houses-is-growing-so-who-wants-them-and-why-83872)
p2, l80 direct quote should give page number
p3, Methods; The paragraph lists results as well as methods. I would also suggest putting these results (i.e. income levels etc) in a table. The section from line 121 to 131 should be at the beginning of the Results section.
p3, l123; is race relevant to a study on tiny houses?
p3, l125; political self-identification seems too broad and undefined to be useful?
p4, l132; should be 'the' summer
p4, l158; seems out of context, perhaps put at the end of the article
p7, l282; it would be good to know for what period this amount refers to (monthly, weekly etc?)
p7, l258; this is confusing, a $35,000 savings on what?
p10, l442 to 444; ensure consistency, use $ OR dollars not both.
p11, l471; why are the words good, easy and practical in quotes and with initial capitals?
General comments:
The paragraphs are very long. Splitting very long paragraphs would improve readability
Could the author/s please define 'good life' as the meaning is unclear
It would have been good to see some more graphical elements, for example, a word cloud of motivators (can be generated by Nvivo)
Use the same terminology throughout; in some places, TH enthusiasts is used, whereas in others, 'tiny-housers' is used. I prefer the former.
Some of the text has strange gaps, p4, l137
Author Response
Response to Reviewer One Comments
We thank the reviewer for the helpful their helpful comments on grammar, style and content. We believe that the feedback helped us produce a better manuscript. In the following section below, we outline the specific changes we have made to the paper.
Point 1: “I” suggest add Shearer, H as a reference, although Australian, she is an active researcher in tiny houses (i.e. Towards a Typology of Tiny Houses, H Shearer, P Burton - Housing, Theory and Society”
Response 1: We thank the reviewer for this reference. We have added it to the manuscript.
Point 2: “l80 direct quote should give page number”
Response 2: We have included the page number for this and all other direct quotes.
Point 3: “Methods; The paragraph lists results as well as methods. I would also suggest putting these results (i.e. income levels etc) in a table. The section from line 121 to 131 should be at the beginning of the Results section”
Response 3: We agree with the reviewer. We created a table and moved the description of the sample to the results section.
Points 4: “is race relevant to a study on tiny houses”.
Response 4: That is a good question and we are not sure we have an answer for this. Previous lifestyle movements (e.g. voluntary simplicity) tended to be a White middle-class / upper-middle class phenomenon. We included race because we wanted to see if the tiny house movement follows the same pattern. We have anecdotal evidence from our (other) research at tiny house festivals that minorities seem to become increasingly interested in tiny houses too … albeit it seems to remain – from what we can tell - a fairly white phenomenon (at least here in the South). Future research will have to shed light into this but we wanted to at least present our findings.
Point 5: “political self-identification seems too broad and undefined to be useful”
Response 5: We included a footnote (footnote 4) in the methods section briefly explaining our rationale. Here is the verbatim.
“Information on the interviewee’s race, religious affiliation, education level, household income and political orientation was collected. Rather than using a pre-made survey item for political orientation, we allowed the participants to self-identify (and coded their responses into a broad conservative, liberal, libertarian and “other” category). We thought that this information could prove useful in providing further insights as whether “ideological leanings” create particular affinities to tiny living. This relationship proved to be difficult to disentangle empirically.”
Point 6: “l132; should be 'the' summer”
Response 6: We fixed this.
Point 7: “p4, l158; seems out of context, perhaps put at the end of the article”
Response 7: We agree with the reviewer and we moved the sentence to the end of the article.
Point 8: “I282; it would be good to know for what period this amount refers to (monthly, weekly etc?)”
Response 8: Good point. We added that information (it’s per month).
Point 9: “p7, l258; this is confusing, a $35,000 savings on what?”
Response 9: We agree that this quote was somewhat confusing. We added clarifying information. The participant (as you can see from the new Table 1) fell into the $50,000 to $74.999 income bracket. What the interviewee meant is that after they have covered their 15k of expenses, there are an extra $35,000 dollars left each year that they can save. We hope that makes more sense.
Point 10: “p10, l442 to 444; ensure consistency, use $ OR dollars not both.”
Response 10: We fixed this problem and are using the $.
Point 11: “p11, l471; why are the words good, easy and practical in quotes and with initial capitals?”
Response 11: This was our attempt to emphasize these words but in order to avoid confusion we have removed the quotation marks.
Point 12: “The paragraphs are very long. Splitting very long paragraphs would improve readability”
Response 12: We agree that some of the paragraphs were too long. We have broken many of the paragraphs (especially in the conclusion) up into shorter paragraphs (while trying to keep the flow of arguments intact).
Point 13: “Could the author/s please define 'good life' as the meaning is unclear”
Response 13: This is a great suggestion. Our views on the Good Life have been strongly influenced by Skidelsky & Skidelsky’s 2012 treatise “How Much is Enough? Money and the Good Life”. New York, Other Press LLC. The authors define the Good Life as “a life that is desirable, or worthy of desire”. While specific conceptions have varied across time and space, Skidelsky & Skidelsky (2012) maintain that there is broad philosophical agreement on what constitute the “basic goods [or elements]” of the Good Life (i.e. health, respect, security, relationships of trust and love). Building on Amartya Sen’s theoretical work on human capabilities (e.g. the capacity to be healthy”), the authors argue that these universal “basic goods … are not just means to, or capabilities for a good life; they are the good life.”
Point 14: “It would have been good to see some more graphical elements, for example, a word cloud of motivators (can be generated by Nvivo)”
Response 14: We thank the reviewer for this suggestion and agree that word clouds help to provide a quick visual overlook of key ideas captured by each theme (elements of the good life). We have created six word clouds (one for four of the five motivators and two for motivator 4: simple living). We hope that these word clouds help to strengthen and highlight the arguments we made in the paper.
Point 15: Use the same terminology throughout; in some places, TH enthusiasts is used, whereas in others, 'tiny-housers' is used. I prefer the former.
Response 15: For the sake of consistency, we are now using TH enthusiasts throughout the paper.
Point 16: “Some of the text has strange gaps, p4, l137”
Response 16: We fixed this problem
Reviewer 2 Report
Excellent paper. There is a real shortage of academic work on the tiny house movement and it's followers despite 20 years of growth and increased acceptance.
I found the quote given in lines 285 through 287 a little confusing. Is there a $15,000 annual savings but total of $35,000 saved since he moved into the tiny house? I'm not sure if it's easy to fix as a direct quote.
On line 295, it is customary to avoid using a number to start a sentence so I would change it from 83% to "Eighty three percent"
On line 360, the quote is "and care take of" her parents. Should this be "caretaker of" her parents or is this how she stated it?
On line 395, the quote states that Bernie is "100 capitalist" and I think that he may have said "100% capitalist"
On line 442, he is quoted as saying "but I don't (them)..." I think something was left out.
Author Response
Response to Reviewer Two Comments
We thank the reviewer for the helpful their helpful comments on grammar, style and content. We believe that the feedback helped us produce a better manuscript. In the following section below, we outline the specific changes we have made to the paper.
Point 1: “I found the quote given in lines 285 through 287 a little confusing. Is there a $15,000 annual savings but total of $35,000 saved since he moved into the tiny house? I'm not sure if it's easy to fix as a direct quote”
Response 1: We agree with the reviewer that the quote sounded confusing the way it was originally written and therefore has been changed. It now reads as follows: “My actual expenses yearly are a little less … than $15,000 so there's a $35,000 savings that I'm just putting in the bank and it's making money for me so I can do things … travel oversees … Give gifts to my family you know. It's just a freeing experience.” The ellipses take the place of the interviewer repeating the same word multiple times.
Point 2: “On line 295, it is customary to avoid using a number to start a sentence so I would change it from 83% to ‘Eighty three percent’”
Response 2: We agree with the reviewer and therefore made the change to reflect the reviewer’s recommendation
Point 3: “On line 360, the quote is ‘and care take of’ her parents. Should this be ‘caretaker of’ her parents or is this how she stated it?”
Response 3: In order to be grammatically correct, it would have to be changed to “caretaker of”, but the way it is currently written are the participants words verbatim. To clear up the issue we have made use of the [sic] to reflect the nature of the participants words.
Point 4: “On line 395, the quote states that Bernie is ‘100 capitalist’ and I think that he may have said ‘100% capitalist’”
Response 4: Upon examining the interview transcript, the reviewer was found to be correct. This error has been corrected and now reads as “100 percent capitalist”.
Point 5: “On line 442, he is quoted as saying ‘but I don't (them)...’ I think something was left out.”
Response 5: Upon a close examination of the transcript it was clear that there were words that were missing which led to an unclear interpretation. This has been fixed and the quote now reads as the following: “I'm an experience person. I have nice things, but I don't love my things … I'd rather spend $2000 dollars on a trip to South America than on a TV … I have a $300TV … I'd rather take that other 1500 dollars and spend it on an experience.”